# Numerical Construction of the Homogenized Strength Criterion for Fiber-Reinforced Composite

**Alexander G. Kolpakov [1,2,\*] and Sergei I. Rakin [2]**

[1] SysAn, A. Nevskogo Str., 12a, 34, 630075 Novosibirsk, Russia
[2] Mathematics Department, Siberian Transport University, Dusi Kovalchuk Str., 191, 630049 Novosibirsk, Russia
[\*] Correspondence: algk@ngs.ru

**Abstract:** In this paper, we investigate whether the strength characteristics of composite materials can be described through the predictions of the homogenization theory concerning local stresses. We establish the homogenized strength criterion (HSC) of composite materials, following the general scheme developed in the homogenization theory. Since the homogenization theory involves solving the so-called periodicity cell problem (PCP), HSC can be constructed in the form of a computer procedure only. We developed the HSC computer program and carried out numerical calculations for fiber-reinforced material. We conclude that HSC can be used to calculate safety zones and the first failure strength criteria (see detailed definitions below). We present numerically calculated safety zones and fracture surfaces for several cases.

**Keywords:** fiber-reinforced composite; homogenization; homogenized strength criterion

## 1. Introduction

The construction of the homogenized strength criteria (HPC) of composite materials attracts the attention of both researchers and engineers [1–10]. HSC is a criterion for the strength of composite components, written in terms of macroscopic SSS. In [2,3], a theoretical scheme for constructing an HSC was proposed. The restriction of the approach [2,3] is that it is a framework scheme that cannot be directly applied to a particular composite. We applied in [1] the theoretical scheme [2,3] to fibrous composites widely used in engineering [10–16]. The approach developed in [1] is briefly presented below (see Formulas (1)–(6) and computation scheme in Section 4). The calculations in [1] are based on the effect of SSS localization [16–21] in a composite with densely packed fibers. The dense packing of fibers in composites is a common but still special case. In addition, calculations based on the SSS localization effect take into account only the principal terms, which gives approximate results. If one wants to have precise strength criteria suitable for general cases, it is necessary to implement the general procedure described in [2,3]. Below, we will explain that the procedure from [2,3] cannot be implemented explicitly, but it is possible to construct a HSC in the form of computer programs. We develop relevant programs and present an example of HSC.

Although the idea of HPC was formulated several decades ago, it has not been fully implemented, primarily due to computational problems. The theory of homogenization in the earliest publications [22–25] discussed both homogenization (also called microscopic, general, and global) characteristics and the relationship between local and homogenized stress–strain state. It has been stated, see [25], for example, that the aforementioned relationship, in perspective, leads to HSC. The calculation of both homogenization characteristics and local stress–strain states in a composite involves solving the so-called periodicity cell problem. To calculate all the homogenized characteristics, it is necessary to solve six periodicity cell problems (according to the number of elements of the stress or strain tensor). When using the theory of homogenization to directly construct a fracture surface

(or similar objects), it is necessary to solve six periodicity cell problems for each value of the homogenized stress/strain. The more or less accurate design of the fracture surface involves taking into account the thousands of values of the homogenized stresses/strains. In the 1960s–1980s, the power of computers was sufficient to solve (often with low accuracy) problems with periodicity cells with simple geometry. A very popular object for numerical analysis in 1960s–1980s were fiber-reinforced composites reinforced with a unidirectional system of fibers. This theme remained actual until now, see, e.g., [26,27]. Frequently, the numerical analysis results for the unidirectional layer were used for theoretical reasoning in order to construct the strength criteria of composites with multidirectional reinforcement.

For a composite with complex geometry, computational difficulties (both related to the power of the computer and problems of theoretical nature) increase drastically. This likely led to a "gap" between research on unidirectional fiber-reinforced composites and complex structure composites. With increasing computer power, researchers are turning their attention to composites with a complex structure. Considerable attention is being paid to the numerical modeling of textile composites, such as knitted and woven, see [28–33] and references in this publications. As a rule, the authors developed numerical methods for calculating a composite subject to a specific SSS. Although the construction of the HSC (the composite calculation taking into account the set of SSSs) is the logical next step, this next step has not been taken in practice. Once again, this step, if it follows the homogenization theory directly, involves a much larger calculation than that required in the solution to specific SSS.

According to the results of a bibliographic search on the Internet, it should be stated that the number of publications devoted to the use of averaging theory for calculating strength is significantly (by an order with guarantee, perhaps two) less than the number of publications devoted to the use of homogenization theory to calculate the average characteristics of a composite. From an engineering point of view, the importance of material characteristics such as Young's modulus and tensile strength seems to be equivalent.

Recently, new papers on the HSCs have appeared, see [34–37]. The recent works are restricted via the construction of certain 2D fragments or sections of HRS failure surfaces. By using our computer program, we computed a "safety zone" and "HSC failure surface" for the planar overall stress–strain state and presented corresponding 3D pictures of the aforementioned objects.

## 2. Homogenization Method as Applied to Composite Reinforced by Systems of Fibers

The homogenization method [22–25] has solved the principal problem of the mechanics of composite materials—the computation of the macroscopic (homogenized) characteristics of composite material. It gave rise to new scientific directions related to homogenized characteristics, for example, the topology design theory [38–40] and auxetic materials [41–45]. In another fundamental problem—the calculation of the strength properties of a composite material—progress was faring worse than in the previously mentioned issue. Especially small progress was made in the construction of HSC for specific composites. As the theoretical scheme of the construction of the HSC of composite material has been known since the 1990s (see [2,3]), we propose that the limited progress in the field is related to computational difficulties. We refer to computational difficulties as both problems with computational resources (hardware and software) and computational methodology.

The problem of composite strength has continuously attracted the attention of the researchers and engineers since the first papers on the strength [2–8] of the composite [46–48]. Many strength criteria have been developed for composite materials. Note that in all the articles mentioned above we find the same basic scheme: typical modes of local SSS are introduced; a connection between local SSS modes and macroscopic SSS is established; and the strength criteria of the composite components are written in terms of macroscopic SSS. This means that the purpose of these articles was the construction of the HSC, even if the term "homogenized strength criterion" was not used at all.

Today, computing resources (hardware and software) are sufficient to implement the methodology from [2,3] with the accuracy required in engineering. So, the problem is methodology. In this work, we construct HSC for a composite reinforced with layers of orthogonal fibers. The constructed HSCs resemble certain well-known strength criteria but do not fully match any of them.

Let us consider a composite reinforced with layers of orthogonal fibers. We will consider this simple reinforcement scheme in order not to clutter up our presentation with unnecessary details. We assume that the layers are parallel to the $Ox_1x_2$-plane and the fibers in the layers are parallel to $Ox_1$ or $Ox_2$-axes (Figure 1). The characteristic dimension $\varepsilon$ of the microstructure of the composite (the radius of the fibers and the distances between the fibers) is assumed to be small compared to the size of the material sample: $\varepsilon << 1$.

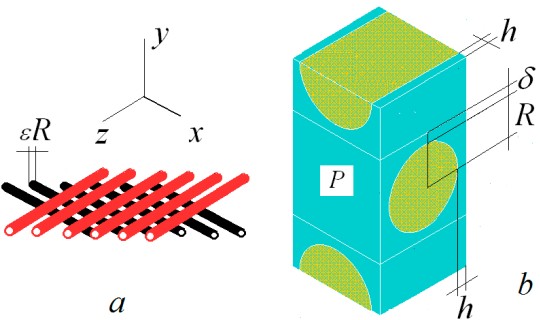

**Figure 1.** Layers of fibers in "slow" variables $\mathbf{x}$ and PC $P$ of composite in the "fast" variables $\mathbf{y}$.

To describe this two-scale material, the "fast" (microscopic) variables $\mathbf{y} = \mathbf{x}/\varepsilon$ are introduced in addition to the "slow" (macroscopic) variables $\mathbf{x}$ [15–17]. Figure 1b displays the periodicity cell (PC) $P = [0, h_1] \times [0, h_2] \times [0, h_3]$ of the composite in the "fast" variables. Denote $\Gamma_i = [0, h_j] \times [0, h_3]$ and $\Gamma_i + h_i\mathbf{e}_i$ ($i = 1, 2$ and $j = 1, 2$) as the opposite lateral faces of PC.

We consider the elasticity problem of composite (it means the fibers and matrix are assumed to be elastic and the connection between the fibers and the matrix is "ideal" [9–11]). The basis for the construction of the strength criterion is the analysis of the local SSS in the composite. A complete analysis of the SSS in the components of the fiber-reinforced composite was carried out in [1] by using the two-scale method, see [15–17]. The main result of [1] is that the local stresses $\sigma_{pq}^{loc}(\mathbf{y})$ in the composite, subjected to the macroscopic strains $\varepsilon_{mn}$, are computed by the formula:

$$\sigma_{pq}^{loc}(\mathbf{y}) = \varepsilon_{mn}(\mathbf{x})a_{pqkl}(\mathbf{y})Z_{k,l}^{mn}(\mathbf{y}) \tag{1}$$

where elastic constants $a_{ijkl}(\mathbf{y}) = a_{ijkl}^F$ in the fibers, $a_{ijkl}(\mathbf{y}) = a_{ijkl}^M$ in the matrix, and the functions $\mathbf{Z}^{\alpha\beta\nu}(\mathbf{y})$ ($\alpha, \beta = 1$ or $3$, $\nu = 0$, $1$) are solutions to the boundary-value problem (4) from [1] (often referred to PCP).

By using the local stresses $\sigma_{pq}^{mn}(\mathbf{y}) = a_{pqkl}(\mathbf{y})Z_{k,l}^{mn}(\mathbf{y})$ corresponding to PCP (4) from [1], we can write (1) as the following:

$$\sigma_{pq}^{loc}(\mathbf{y}) = \varepsilon_{mn}(\mathbf{x})\sigma_{pq}^{mn}(\mathbf{y}) \tag{2}$$

After the functions $\mathbf{Z}^{mn}(\mathbf{y})$ have been computed, the local stresses $\sigma_{pq}^{loc}(\mathbf{y})$ in the composite are computed following (1) or (2). The effective (homogenized) constants $A_{pqmn}$ of the composite are given by the formulas [15–17]

$$A_{pqmn} = \frac{1}{mesP} \int_P a_{pqkl}(\mathbf{y})Z_{k,l}^{mn}(\mathbf{y})d\mathbf{y} \tag{3}$$

Hereafter, $mesP$ means volume in 3D cases and square in 2D cases.

The homogenized stress and strains are connected by the homogenized Hook's low $\sigma_{kl} = A_{klmn}\varepsilon_{mn}$.

## 3. The Strength of the Composite

Our goal is to obtain the HSC of the composite. We call the homogenized strength criterion (HSC) the strength criterion of composite components (fibers and matrix), written in terms of homogenized strains $\varepsilon_{mn}$ or homogenized stresses $\sigma_{kl}$. We assume that the strength criterion of the material of the fibers and the binder may be written in the form $f(\mathbf{y}, \sigma_{pq}^{loc}) \leq \sigma^*(\mathbf{y})$, where

$$\sigma^*(\mathbf{y}) = \left\{ \begin{array}{l} \sigma_F^* \text{ in fiber} \\ \sigma_M^* \text{ in binder} \end{array} \right.,$$

$$f(\mathbf{y}, \sigma_{pq}^{loc}) = \left\{ \begin{array}{l} f_F(\sigma_{pq}^{loc}) \text{ in fiber} \\ f_M(\sigma_{pq}^{loc}) \text{ in binder} \end{array} \right. \tag{4}$$

In (4), $\sigma_F^*$ is the strength limit of the fibers and $\sigma_M^*$ is the strength limit of the matrix. The following condition

$$F(\varepsilon_{mn}) = \max_{\mathbf{y} \in P} \frac{f(\mathbf{y}, \varepsilon_{mn} a_{pqkl}(\mathbf{y}) Z_{k,l}^{mn}(\mathbf{y}))}{\sigma^*(\mathbf{y})} < 1 \tag{5}$$

ensures no damage at all the points of PC $P$. For this reason, we call the strains $\{\varepsilon_{mn}: F(\varepsilon_{mn}) < 1\}$ the safety zone $V$.

The destruction of composite starts when the condition $F(\varepsilon_{mn}) = 1$ is satisfied and occurs at point(s) $\mathbf{y}_0 \in P$, at which the maximum in (5) reaches values 1:

$$\frac{f(\mathbf{y}_0, \varepsilon_{mn} a_{pqkl}(\mathbf{y}_0) Z_{k,l}^{mn}(\mathbf{y}_0))}{\sigma^*(\mathbf{y}_0)} = 1 \tag{6}$$

The destruction of composites is a complex multistage process. This thesis was formulated in the initial works on composite materials [18,19] and is still accepted, see, for example, [20,21,47,48]. Therefore, when speaking about the strength or fracture of a composite, it is necessary to indicate which stage of fracture is being discussed. HSC (6) $F(\varepsilon_{mn}) = 1$ is the "first crack" criterion and $\mathbf{y}_0$ indicates the "weakest element" of the composite. The fulfillment of Equation (6) $F(\varepsilon_{mn}) = 1$ does not mean that the composition breaks up into separate parts. It is not excluded that the composite will not even lose its load-bearing capacity. In any case, if the condition $F(\varepsilon_{mn}) = 1$ is met, the "first cracks" appear in numerous PCs, which means that damage to the composite will be massive, and this stage should be singled out as the specific stage of the destruction process. We say that deformations $\{\varepsilon_{mn}: F(\varepsilon_{mn}) = 1\}$ form a fracture surface $S$.

The condition (5) $F(\varepsilon_{mn}) < 1$ guarantees that the composite does not fail when strains $\varepsilon_{mn}$ are applied to the composite. For this reason, we say that deformations $\{\varepsilon_{mn}: F(\varepsilon_{mn}) < 1\}$ form a safety zone $V$.

The safety zone $V$ and the destruction surface $S$ belong to $R^3$. The destruction surface $S$ is the boundary of the safety zone $V$.

Formulas (5) and (6) can be written in terms of homogenized stresses $\sigma_{mn}$ if the homogenized strains $\varepsilon_{mn}$ are expressed through $\sigma_{mn}$ by using the homogenized Hook's low $\varepsilon_{mn} = A_{mnkl}^{-1}\sigma_{kl}$ and then this expression is substituted to (5) and (6).

## 4. Construction of HSC in the Form of a Computer Program

The construction of HSC in [1] was based on the SSS localization effect [34–38] in the high-contrast composite. In [1], we use the SSSs localized in different domains for the different deformation modes. As a rule, this is not the case. In addition, calculations based on the SSS localization effect are approximate, since only leading terms are taken

into account. To construct a universal and accurate HSC, it is necessary to implement the general procedure described above. The construction of HSC, in general or special cases, assumes the computation of the local stresses $\sigma_{pq}^{loc}(\mathbf{y})$ in the PC. In the general cases, these computations can be performed only numerically. Let us describe our proposed computational scheme. By virtue of Formula (1), the construction of the GSP can be divided into two computational blocks. The first is for PCP solution. This block is responsible for the microstructure of the composite. The other is for calculating HSC value. The detailed description of the blocks is given below.

---

**Block of micro-structural analysis:** solve PCP (4) for the unit value strains $\varepsilon_{mn} = \delta_{mn}$, compute the local stresses $\sigma_{pq}^{mn}(\mathbf{y}) = a_{pqkl}(\mathbf{y})Z_{k,l}^{mn}(\mathbf{y})$ in all finite elements, and save $\sigma_{pq}^{mn}(\mathbf{y})$ into files.

---

$\downarrow \sigma_{pq}^{mn}(\mathbf{y})$-files transfer

---

**Block HSC:** compute the local stresses $\sigma_{pq}^{loc}(\mathbf{y})$ in all finite elements for the given macroscopic strains $\varepsilon_{mn}$ following Formula (1) and check the condition (5).

---

In order to determine the failure surface, it is necessary to replace "check condition (5)" in the HSC block with "select macroscopic strains $\varepsilon_{mn}$ that satisfy the equation $F(\varepsilon_{mn}) = 1$ (and draw these points if one wants for a visualization of the fracture surface)". The calculations mentioned in the first block can be performed using the ANSYS (or similar) FEM software. The calculations mentioned in the second block can be performed using a program developed for this case (the authors developed program in C). The main component of our computer program, except the file exchange procedure, is the procedure for numerical solution to Equation (6). The program also includes a graphics procedure, which can be used for illustrative purposes only. Another version of the second program computes the HSC value (5) and decides whether the homogenized stresses/strains belong to the safety zone. The integration of HSC with ANSYS (or similar) FEM software is a separate problem.

In the present configuration, the computational programs are independent. The first block program accounts for the local (material and geometrical) characteristics of the composite. It does not depend on the macroscopic strains $\varepsilon_{mn}$. The second block program—HSC itself—computes the strength criterion value for prescribed macroscopic $\varepsilon_{mn}$. This program is the same for any composite material. The information about the specific structure of the composite plate is input into this program in the form of the $\sigma_{pq}^{mn}(\mathbf{y})$-files.

The computations mentioned in the first block are time-consuming (from a few minutes to an hour) due to numerical solution to the 3D elasticity problem. The HCS procedure is fast (a few seconds). The failure surface construction procedure is time-consuming due to the many repetitions of the fast HCS procedure.

In our calculations, we used the ANSYS FEM software to solve the linear elasticity problem. For this reason, we did not pay much attention to the convergence and stability of numerical procedures (ANSYS FEM computational procedures satisfy the necessary requirements for an accurate solution of a linear problem of elasticity theory). We adopted the FE mesh, which twice refined the changes solution to less than 5%.

CPU 3 GHz and 1 GB of computer RAM was enough to solve problems with periodicity cells in the "Microstructural Analysis Unit" in 1–10 min. The main problem in the "Block of microstructural analysis" was the generation of a periodic FE grid (the insufficiency of ANSYS in this area is known [49]). Executing a C program in the "HSR Block" in HSR mode takes a few seconds. The execution of program C in the "build security zone" mode was time-consuming and took between 10 and 30 min. This was due to the repeated repetition of the calculations of the "VSM block" in the "Construction of a security zone" mode.

We do not discuss specific issues here, such as interaction between Windows and DOS simulators and the like, which are specific problems associated with the software used by the authors.

## 5. An Example. Construction of the Failure Surfaces and the Safety Zones

We construct the HSC for the fiber-reinforced material considered in [1]. The material parameters of the fibers and the matrix used in our computations are presented in Table 1. These elastic characteristics correspond to the carbon/epoxy composite.

**Table 1.** The material parameters of the fibers and the matrix used in our computations.

|  | Young's Modulus GPa | Poisson's Ratio | Strength Limit Pa |
| --- | --- | --- | --- |
| Fibers | $E_F = 170$ GPa | $\nu_F = 0.3$ | $\sigma_F^* = 1.5 \cdot 10^9$ |
| Matrix | $E_M = 2$ GPa | $\nu_M = 0.36$ | $\sigma_M^* = 60 \cdot 10^6$ |

The dimensions of the components of the composite are the following (Figure 1): $h_1 = 1.1$, $h_2 = 3$, $h_3 = 1.1$, $h = 0.1$, $\delta = 0.1$, and $R = 0.45$, see Figure 1. These values are included in the non-dimensional "fast" variables **y**. The corresponding dimensional values are computed by multiplying by $\varepsilon$. For carbon fibers, $\varepsilon$ varies from 5 to 20 µm ($5 - 20 \cdot 10^{-6}$ m).

The solution to PCP and $\sigma_{pq}^{mn}$ was obtained by using ANSYS FEM software. In computation, finite element SOLID185 was used and the number of finite elements was about 3000.

In our computation, we accept the von Mises strength criterion for the fibers and the binder. In this case, $f(\sigma_{mn}) = \sqrt{\frac{3}{2} s_{ij} s_{ij}}$, where $s_{ij}$ is the deviator of the local stress tensor $\sigma_{mn}$.

In general cases, the failure surface $\{\varepsilon_{mn} : F(\varepsilon_{mn}) = 1\}$ and the safety zone $\{\varepsilon_{mn} : F(\varepsilon_{mn}) < 1\}$ are 6D objects and cannot be visualized in an appropriate way. In the case of macroscopic in-plane deformation, the problem depends on the three macroscopic strains and the corresponding objects may be visualized. For this reason, we consider the macroscopic in-plane deformation in the $Ox_1x_3$ plane. In this case, $\varepsilon_{i3} = 0$ ($i = 1, 2, 3$).

The periodicity cell $P$ is formed of fibers $F$ and matrix $M$: $P = F \cup M$. The strength conditions for the fibers and the matrix separately) take the following form:

$$B_F(\varepsilon_{\alpha\beta}) \leq 1, B_M(\varepsilon_{\alpha\beta}) \leq 1 \tag{7}$$

where $(\varepsilon_{\alpha\beta}) = (\varepsilon_{11}, \varepsilon_{22}, \varepsilon_{12})$, $\alpha, \beta = 1, 3$, and

$$B_F(\varepsilon_{\alpha\beta}) = \max_{\mathbf{y} \in F} \frac{f_F(\varepsilon_{\alpha\beta} a_{pqkl}^F Z_{k,l}^{\alpha\beta}(\mathbf{y}))}{\sigma_F^*}, B_M(\varepsilon_{\alpha\beta}) = \max_{\mathbf{y} \in M} \frac{f_M(\varepsilon_{\alpha\beta} a_{pqkl}^M Z_{k,l}^{\alpha\beta}(\mathbf{y}))}{\sigma_M^*} \tag{8}$$

The equation $B_F(\varepsilon_{\alpha\beta}) = 1$ introduces the failure surface $S_F$ for the fibers and $B_M(\varepsilon_{\alpha\beta}) = 1$ introduces the failure surface $S_M$ for the matrix. The destruction of the fiber starts at the point(s) $\mathbf{y}_{0F} \in F$ at which the first maximum in (8) reaches values 1. The destruction of the matrix starts at the point(s) $\mathbf{y}_{0M} \in M$, at which the second maximum in (8) reaches values 1.

The failure surface $S$ for the periodicity cell (i.e., for composite as a whole) is determined as the $\{\varepsilon_{mn} : \max(B_F(\varepsilon_{\alpha\beta}), B_M(\varepsilon_{\alpha\beta})) = 1\}$. This definition reflects the fact that destruction may start both in the fibers and in the matrix.

The failure surface $S$ may be determined in another equivalent way. $V_F$, $V_M$, and $V$ are the safety zones for the fibers, matrix, and the composite as whole. The domain $V$ is determined by the inequality max ($B_F(\varepsilon_{\alpha\beta})$ and $B_M(\varepsilon_{\alpha\beta})$) $\leq 1$ is the intersection of the domains $V_F$ and $V_M$ determined by the inequalities (7). The failure surface $S$ is the boundary of the domain $V$.

Figure 2 displays the failure surfaces $S_M$ and $S_F$. The safety surfaces are colored in red/blue and pink-yellow for the best visibility of their 3D geometry. In Figure 2a,b, the half of the failure surfaces $S_M$ and $S_F$ are displayed. The remaining parts of the failure surfaces are the mirror reflections of the displayed fragments with respect to the $O\varepsilon_{xx}\varepsilon_{zz}$-plane. The mark 0.01 in the figures corresponds to 1% tension/shift strains. In Figure 2, the failure

surface $S_M$ and a fragment of the failure surface $S_F$ are displayed. The surface $S_F$ is long in the $O\varepsilon_{13}$-direction. This means that the fibers are strong against the macroscopic shift in the $Oxz$-plane. This is in agreement with the "scissor-like" deformation of the fibers [1].

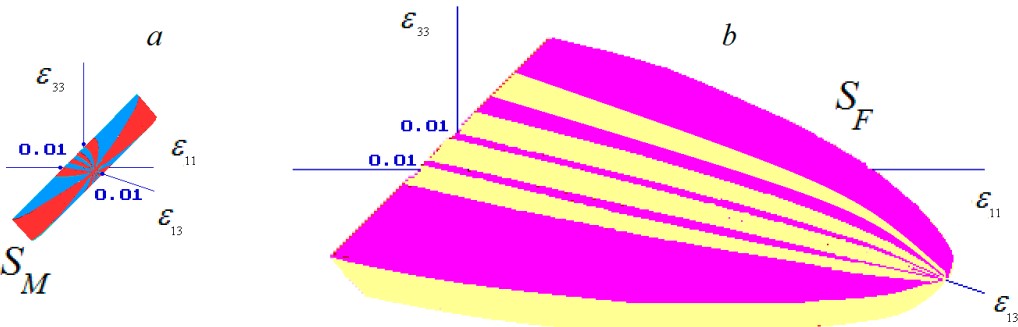

**Figure 2.** Failure surfaces: (**a**)—for the matrix, (**b**)—for the fibers.

The interactions of the failure surfaces with the coordinate planes provide us with additional information concerning the geometry of the failure surfaces. There are three coordinate planes: $\varepsilon_{11} = 0$, $\varepsilon_{33} = 0$, and $\varepsilon_{13} = 0$. Figure 3 displays the failure surfaces $S_M$ and $S_F$ for zero shift ($\varepsilon_{13} = 0$) and arbitrary axial tensions $\varepsilon_{33}$ and $\varepsilon_{13}$. The planar failure surface $S_M$ is colored in yellow for the best visibility. The fracture surface $S_M$ is entirely contained within the fracture surface $S_F$ (it is also seen in Figure 2). This means that the weakest element of the composite is matrix. At the same time, $S_M$ and $S_F$ are placed at a small distance one from another. This means that carbon fibers and epoxy matrix are nearly equal in strength as the components of the composite. Note that the strengths of carbon fibers and epoxy individually are very different, see Table 1.

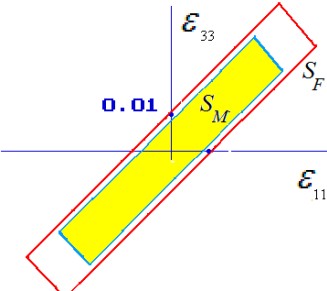

**Figure 3.** Failure surface for matrix and for fiber for the axial tension modes.

Figure 4a displays the failure surfaces for zero axial stress $\varepsilon_{33}$ and arbitrary axial stress $\varepsilon_{11}$ and shift $\varepsilon_{13}$. Figure 4c displays the failure surfaces for the zero axial stress $\varepsilon_{11}$ and the arbitrary axial stress $\varepsilon_{33}$ and shift $\varepsilon_{13}$. The central fragments of Figure 4a,c are enlarged. In both cases, the matrix is the weakest element of the composite. In Figure 4, we see that the strengths of the fibers and the matrix are similar against the macroscopic axial tension and rather different against the macroscopic shift.

We demonstrate the way in which the change of characteristics of components of the composite changes the HSC. The strength limit of epoxy may change from $30 \cdot 10^6$ to $90 \cdot 10^6$ [50]. Figure 5 displays the interactions of the failure surfaces with the coordinate planes when the epoxy strength limit is changed to $\sigma_M^* = 85 \cdot 10^6$ Pa (the other characteristics are the same). Then, the failure surface $S_F$ is placed inside the failure surface $S_M$ (see Figure 5a for the axial tension mode). This means that the fibers become the weakest element of the composite. In the tension-shift case, the failure surface $S_M$ goes beyond the failure surfaces $S_F$, Figure 5b. The failure surface of the composite as a whole consists of the fragments of the failure surfaces of the fibers and the matrix, see Figure 5c. When

strains $(\varepsilon_{11}, \varepsilon_{33})$ belong to $AB$ or $DA$ in Figure 5c, the fibers are destroyed. When the strains $(\varepsilon_{11}, \varepsilon_{33})$ belong to $BC$ or $CD$, the matrix is destroyed.

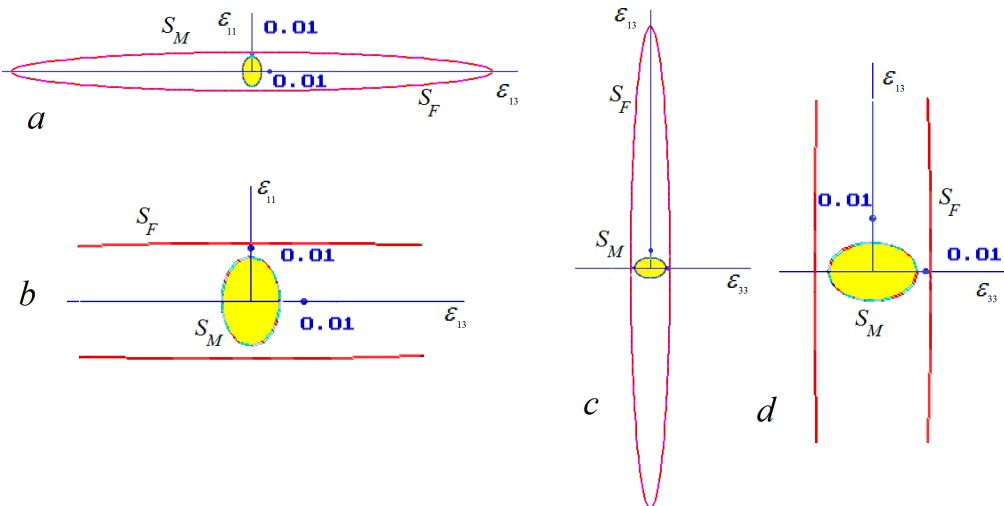

**Figure 4.** Failure surface for the tension-shift modes: (**a**)—in plane $O\varepsilon_{11}\varepsilon_{13}$, (**b**)—enlarged, and (**c**)—in plane $O\varepsilon_{13}\varepsilon_{33}$, (**d**)—enlarged.

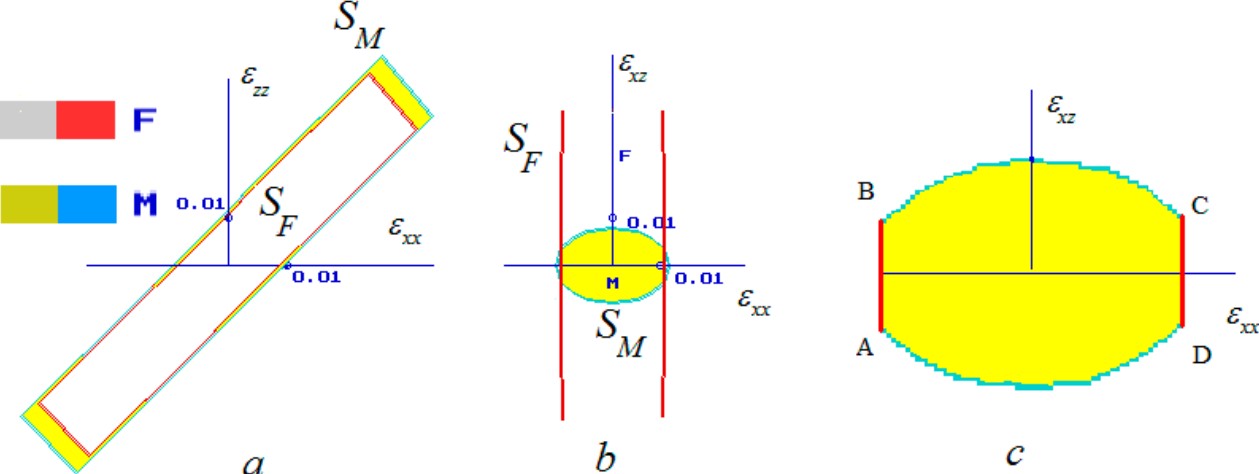

**Figure 5.** Failure surface: (**a**)—for the tension modes; (**b**)—for tension-shift mode; (**c**)—for composite as a whole (the central part of Figure (**b**), enlarged).

Figure 6 displays the 3D failure surfaces for composite as a whole when $\sigma_M^* = 85 \cdot 10^6$ Pa. It consists of the fragments of the failure surface for the fibers and matrix.

Although we use the von Mises strength criterion for the components of the composite, Figures 2–6 demonstrate no similarity with the von Mises criterium at the macrolevel. The proportions of the right angle sides in Figure 6 are 1:4 for the matrix and approximately 1:6 for the fibers. One can treat the long length of the failure surfaces along the line $\varepsilon_{11} + \varepsilon_{33} = 0$ as a residual form of the hydrostatic axis.

The failure surface of the composite is the intersection of the domains $V_F$ and $V_M$ (8). Usually, intersection $V_F \cap V_M$ has a non-smooth boundary, even if $S_F$ and $S_M$ are smooth surfaces. This kind of non-smoothness results from the change of the failure mode from the "matrix failure" mode to the "fiber failure" mode (or vice versa) when the macroscopic SSS changes.

Figures 3–5 show that the fracture surface $S_F$ of the fibers can also be non-smooth. Thus, the fracture surfaces of fibers as an element of a composite are not directly related to the strength criteria of fiber materials. In the example above, we used the von Mises

strength criteria for the material of the fibers. The well-known von Mises fracture surface is smooth and shows no similarity with $S_F$.

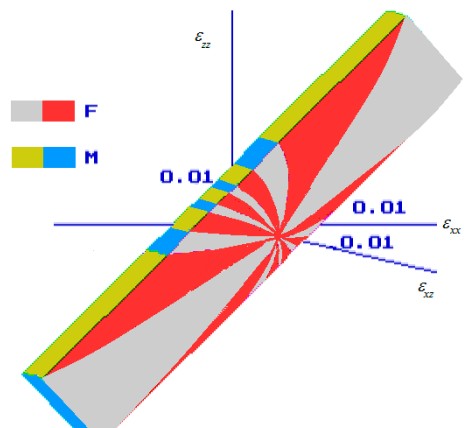

**Figure 6.** The failure surfaces when the epoxy strength limit $\sigma_M^* = 85 \cdot 10^6$ Pa.

## 6. Conclusions

We present an implementation of HSC for fiber-reinforced material based on the homogenization theory. In general, an HSC for a fiber-reinforced composite can be developed as a computer program. The original HSC-authoring program was developed in C/C++. The program uses auxiliary data calculated using the ANSYS FEM program. The developed HSC program causes it to determine the "safety zone" and the "the first crack failure surface", to identify the "weakest" component of the composite (fiber or matrix), as well as the weakest point in the composite.

HSC (at least in its current form) is not designed to analyze progressive damage or the complete failure of a composite. Considering the reliability of ANSYS FEM (or similar FEM software) for solving problems in the theory of elasticity, the authors consider it possible to use HSC to predict the "safety zone" for composites.

**Author Contributions:** A.G.K. methodology; S.I.R. computations, programming. All authors have read and agreed to the published version of the manuscript.

**Funding:** This research received no external funding.

**Data Availability Statement:** All the data are included into paper.

**Conflicts of Interest:** The authors declare no conflict of interest.

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
