# Peer review of "Numerical Construction of the Homogenized Strength Criterion for Fiber-Reinforced Composite"

_jcs, doi:10.3390/jcs7040145_

Round 1

Reviewer 1 Report (Previous Reviewer 1)

The presentation has been corrected improved. 

I think that the paper contains a new interesting result and can be published in J. Compos. Sci. 

Author Response

Thanks

Reviewer 2 Report (Previous Reviewer 2)

The reviewer recommends rejecting the article and resubmitting it after thoroughly revising the content, using experimental validation and real-world cases to validate the numerical models.

Author Response

The reviewer recommends rejecting the article and resubmitting it after thoroughly revising the content, using experimental validation and real-world cases to validate the numerical models.

REPLY: The purpose of the article is "Numerical construction of a homogenized criterion for the strength of a fibrous composite", which is a theoretical/computational problem in the homogenization theory, which does not require experimental or real-world justification.

We definitely say that we do not pretend to solve many other interesting and important problems in the mechanics of composites.

In addition to the numerical construction of the HSR, we conclude that HSR describes the "safety zone" of composite and the violation of the HSR leads to failure(s) in some points of composite. These conclusions are based on the elasticity theory and accepted strength theories.

We ask do not treat this paper in any expanded sense.

Reviewer 3 Report (Previous Reviewer 3)

No discussion with open literature. The conclusions can be questionable. 

Author Response

No discussion with open literature. The conclusions can be questionable. 

REPLY: We have added references [22-26] directly discussed the application of the homogenization method to the problem of strength. The existing papers on the homogenized strength criterion did not present “the homogenized failure surfaces” or other objects suitable for the comparison. Taking into account that the HSC problem was formulated several decades ago, it seems strange, but it is. The conclusions section is rewritten.

Round 2

Reviewer 2 Report (Previous Reviewer 2)

In the introduction, the authors should provide a more comprehensive review of the existing literature on homogenized strength criteria for fiber-reinforced composites. Provide more context and background information: While the paper does provide a brief overview of the method presented in [1], it may be helpful to provide additional context and background information on the topic of fiber-reinforced composites and the need for a homogenized strength criterion.

The authors should provide more details on the numerical implementation of the method and the computer program used to construct the HSC. Specifically, they should provide information on the convergence criteria used in the numerical computations, as well as the computational resources required for the calculations.

In the section on results, the authors should include more discussion on the implications of their findings. Specifically, they should discuss how the visualizations of the failure surface for the low-dimensional cases can be used to predict failure behavior in actual engineering applications.

The authors should consider including more detailed descriptions of the mathematical formulations used in the paper. This will help readers with a strong mathematical background better understand the methodology and results.

Finally, the authors should consider providing a conclusion that summarizes the main findings of the paper and discusses potential future research directions. This will help readers better understand the significance and implications of their work.

Author Response

2 In the introduction, the authors should provide a more comprehensive review of the existing literature on homogenized strength criteria for fiber-reinforced composites. Provide more context and background information: While the paper does provide a brief overview of the method presented in [1], it may be helpful to provide additional context and background information on the topic of fiber-reinforced composites and the need for a homogenized strength criterion.

REPLAY: The approach developed in [1] is briefly presented below (see formulas (1)-(6) and computation scheme in Section 4). We also add literature

The authors should provide more details on the numerical implementation of the method and the computer program used to construct the HSC. Specifically, they should provide information on the convergence criteria used in the numerical computations, as well as the computational resources required for the calculations.

REPLAY: The information is added.

In the section on results, the authors should include more discussion on the implications of their findings. Specifically, they should discuss how the visualizations of the failure surface for the low-dimensional cases can be used to predict failure behavior in actual engineering applications.

REPLAY: To the best knowledgr of the authors, HSCs wer not constructed eariler. In this paper, the authors want give first images of HRS.

 The authors should consider including more detailed descriptions of the mathematical formulations used in the paper. This will help readers with a strong mathematical background better understand the methodology and results.

REPLAY: We agree that the homogenization theory is a sophysticated mathematical theory. The aim of this paper is realization of theoretical idea of HSC. We do not want the sophysticated mathematics musks our computational results.

Finally, the authors should consider providing a conclusion that summarizes the main findings of the paper and discusses potential future research directions. This will help readers better understand the significance and implications of their work.

REPLAY: In this paper, we want present a realization of HSC from the homogenization theory. We consider thi paper as a start of the discussion.

Reviewer 3 Report (Previous Reviewer 3)

Although the discussion is short, the authors justify and tried to add open literature.

Author Response

3Although the discussion is short, the authors justify and tried to add open literature

REPLAY: We add new references more

Round 3

Reviewer 2 Report (Previous Reviewer 2)

accept in present form

This manuscript is a resubmission of an earlier submission. The following is a list of the peer review reports and author responses from that submission.

Round 1

Reviewer 1 Report

The paper “Numerical construction of the homogenized strength criterion for fiber-reinforced composite” by Alexander G. Kolpakov, Sergey I. Rakin is devoted to the elastic interaction of two arrays of fibers displayed in Fig. 1 in the interface zone. The combination of asymptotic analysis based on the boundary layer approximation is developed and the FEM computations are used to discuss the homogenized strength and the delaminated strength criteria. Though the considered problem attracted the attention of many engineers, it was investigated by the naive reiterated homogenization when two unidirectional arrays were first homogenized and the interaction of the obtained homogenized media was investigated. Various ad-hoc modifications were applied. Nevertheless, the previous estimations can be used only for sufficiently large distances between the arrays.

I think that this paper is one of the first serious attempts to solve the problem for interacting arrays.

I have the following minor remarks concerning the presentation. The references [r10-r12] on line 34 and [t13-t15] on line 37 should be arranged. The brackets on lines 159 and 163 are separated from the considered sets. References on line 270 should be unified.

I suppose that the authors should carefully read the paper and improve its presentation. I do not say about the numerical results and the general theoretical conception. I mean many simple bugs.     

Author Response

We thank the Reviewers for criticism on the defects in the paper (sorry!) and suggestions for improving the paper. We fully agree with criticism and suggestions. We corrected defects. Note that the paper considers two characteristics of the composite: the safety zone and the destruction of the "first crack". We agree that these characteristics are very conservative failure conditions. We discuss this point in the paper e. We did not take into account progressive damage. Using the developed approach, it is possible to analyze the nonlinear/plastic deformation of the composite. It seems that this approach can account for microcracks. These are topics for further research and articles. We understand the importance of comments regarding the progressive development of cracks in the composite, but are not ready to discuss these issues in this article.

The massive corrections are marked in yellow.

English was checked once more with on-line editor and our English colleague.

1

The references [r10-r12] on line 34 and [t13-t15] on line 37 should be arranged. It is done

The brackets on lines 159 and 163 are separated from the considered sets. Thanks. Probably, this is WORD to PDF problem. The original WORD text did not contains these defects.

References on line 270 should be unified. The reference list is formatted

Reviewer 2 Report

While the research work presents a solid analysis and discussion of the findings, the addition of an experimental component would add depth and validity to the study. Incorporating real-life applications and testing of the theories would greatly enhance the overall impact of the research

Author Response

We thank the Reviewers for criticism on the defects in the paper (sorry!) and suggestions for improving the paper. We fully agree with criticism and suggestions. We corrected defects. Note that the paper considers two characteristics of the composite: the safety zone and the destruction of the "first crack". We agree that these characteristics are very conservative failure conditions. We discuss this point in the paper e. We did not take into account progressive damage. Using the developed approach, it is possible to analyze the nonlinear/plastic deformation of the composite. It seems that this approach can account for microcracks. These are topics for further research and articles. We understand the importance of comments regarding the progressive development of cracks in the composite, but are not ready to discuss these issues in this article.

The massive corrections are marked in yellow.

English was checked once more with on-line editor and our English colleague.

2

While the research work presents a solid analysis and discussion of the findings, the addition of an experimental component would add depth and validity to the study. Incorporating real-life applications and testing of the theories would greatly enhance the overall impact of the research

We completele agree with the reviewer. These research research programm assumes numerous papers. We will keep in mind these recommendations. Sorry, we cannot realize this progremm in this paper

Reviewer 3 Report

1References in the abstract are unusual. The abstract must present the conclusions obtained in the study.

2Line 34, 37… [r10-r12]; [t13-t15]. What do the letters before the reference number mean? t-topology; a-auxetic,…but, what is the need?

3 References are not all formatted in the same way.

4 Line 46 – [2-] Missing references?

5 References should be numbered according to the order they appear in the text.

6  Line 82 – replace 0,1 by 0.1.

7 Line 99 – What are the main failure modes of composite materials? Is there interaction between them? What is the failure that appears first?

8  What is the need to use two different software? It is not clear on paper.

9   How many elements and nodes were used? Type of elements?

1 No discussion with open literature. The conclusions can be questionable. The results must be presented, discussed and justified.

Author Response

We thank the Reviewers for criticism on the defects in the paper (sorry!) and suggestions for improving the paper. We fully agree with criticism and suggestions. We corrected defects. Note that the paper considers two characteristics of the composite: the safety zone and the destruction of the "first crack". We agree that these characteristics are very conservative failure conditions. We discuss this point in the paper e. We did not take into account progressive damage. Using the developed approach, it is possible to analyze the nonlinear/plastic deformation of the composite. It seems that this approach can account for microcracks. These are topics for further research and articles. We understand the importance of comments regarding the progressive development of cracks in the composite, but are not ready to discuss these issues in this article.

The massive corrections are marked in yellow.

English was checked once more with on-line editor and our English colleague.

3

References in the abstract are unusual. The abstract must present the conclusions obtained in the study.

2Line 34, 37… [r10-r12]; [t13-t15]. What do the letters before the reference number mean? t-topology; a-auxetic,…but, what is the need? Thanks. Corrections are done

3 References are not all formatted in the same way. The list of references is formatted

4 Line 46 – [2-] Missing references? It is corrected

5 References should be numbered according to the order they appear in the text.

6  Line 82 – replace 0,1 by 0.1. Index \nu takes values 0 or 1.

7 Line 99 – What are the main failure modes of composite materials? Is there interaction between them? What is the failure that appears first? We use another approach to failure. The first zone is the safety zone, where there are no failures. After, our approach identifies the “first crack(s)” failure. Our approach identifies the interaction between the safety zone and the first crack zone. As far as the following process, up to separation of the structure into parts, our approach (at least in the present form) keeps silence.

8  What is the need to use two different software? It is not clear on paper. ANSYS is very good tool to solve the classical engineering problems. We understand, ANSYS realizes known methods. If problem is new, it is heavy to use ANSYS. This is our specific experience, we do not criticize ANSYS or similar FEM tools. ANSYS is great support in our research.

9   How many elements and nodes were used? Type of elements? This information is added

10 No discussion with open literature. The conclusions can be questionable. The results must be presented, discussed and justified. We completely agree with the reviewer. We only note that the principal idea of the homogenization strength criterion is old, but complete realization of this idea starts recently (these is discussed in the paper). There are huge number of the strength of the composite, but we had a problem to find a paper to compare. Underline that the problem is clear formulated – to realize the homogenized strength criterion following to the (classical, now) the homogenization theory. 

Reviewer 4 Report

The submitted article considers the development of a homogenized strength criterion for fiber reinforced composites. The authors develop the strength criterion based on micromechanical considerations of the composite microstructure using representative volume elements (RVE). A strength criterion based on the von-Mises equivalent stress is used in both, matrix and fibers in order to define a strength criterion of the constituents. A two-step numerical approach is used in order to derive a homogenized strength criterion from the local fields within the RVE.

The article has several flaws regarding grammar and content. First of all, a throughout correction regarding English spelling and grammar is required. The text is not easy to read such that it is hard to follow the explanation of the authors. Within the text, the typesetting and figures have to be reworked, for instance:

·        Section 2: The references have letters in front of the numbers

·        Lines 60-61: Typesetting of plane indices

·        Line 78: The indices of the variable Z change from 4 to 3

·        Figure 2c is missing

·        Figure 4-6: Explanatory legend is missing

Moreover, variables and abbreviations are not introduced properly. E.g. PCP was never introduced, mesP in equation 2 is not defined.

Besides the textual flaws, there are also several issues regarding content:

·        To the reviewers understanding, the composite is assumed to fail if one material point/element fails. This assumption is very conservative since it is known that cracks form within a composite (especially thermosets) without the total failure of the composite.

·        The material laws used for fiber and matrix are not introduced. Only the elastic modulus and Poisson’s ratio are given.

·        Assuming the materials are considered as linear elastic, the proposed approach does not consider the influence of the evolution of damage within the constituents. E.g. the failure of a material point does not influence its neighborhood. Thus, the proposed method consists of a simple (linear?) calculation with a subsequent calculation of the von-Mises stress and application of the maximum stress criterion.

·        The developed “computer program” is not outlined in detail. The provided information is very broad. Since this a one of the main novelties of the article, the reviewer considers a more in detail discussion to be appropriate.

Considering the exemplary named flaws of the article, the reviewer recommends to reject the article and to resubmit the article after thoroughly overworking the grammar and contents.

Author Response

We thank the Reviewers for criticism on the defects in the paper (sorry!) and suggestions for improving the paper. We fully agree with criticism and suggestions. We corrected defects. Note that the paper considers two characteristics of the composite: the safety zone and the destruction of the "first crack". We agree that these characteristics are very conservative failure conditions. We discuss this point in the paper e. We did not take into account progressive damage. Using the developed approach, it is possible to analyze the nonlinear/plastic deformation of the composite. It seems that this approach can account for microcracks. These are topics for further research and articles. We understand the importance of comments regarding the progressive development of cracks in the composite, but are not ready to discuss these issues in this article.

The massive corrections are marked in yellow.

English was checked once more with on-line editor and our English colleague.

4

The submitted article considers the development of a homogenized strength criterion for fiber reinforced composites. The authors develop the strength criterion based on micromechanical considerations of the composite microstructure using representative volume elements (RVE). A strength criterion based on the von-Mises equivalent stress is used in both, matrix and fibers in order to define a strength criterion of the constituents. A two-step numerical approach is used in order to derive a homogenized strength criterion from the local fields within the RVE.

The article has several flaws regarding grammar and content. First of all, a throughout correction regarding English spelling and grammar is required. The text is not easy to read such that it is hard to follow the explanation of the authors. Within the text, the typesetting and figures have to be reworked, for instance:

  • Section 2: The references have letters in front of the numbers It is corrected
  • Lines 60-61: Typesetting of plane indices It is result of WORD to PDF convertation. No this defect in the original WORD text. Sorry.
  • Line 78: The indices of the variable Z change from 4 to 3 Comments in the text
  • Figure 2c is missing Corrected
  • Figure 4-6: Explanatory legend is missing Corrected

Moreover, variables and abbreviations are not introduced properly. E.g. PCP was never introduced, mesP in equation 2 is not defined.

Besides the textual flaws, there are also several issues regarding content:

  • To the reviewers understanding, the composite is assumed to fail if one material point/element fails. This assumption is very conservative since it is known that cracks form within a composite (especially thermosets) without the total failure of the composite.

We introduce the notion of the safety zone and the zone of the first crack(s). We describe these zones in detalis. We understand that waterial may works after it leavs the the safety and the first crack(s) zone. We agree, that our criteria are conservative. Our position is that is is necassary to clear formulat the meaning of every criteria.

  • The material laws used for fiber and matrix are not introduced. Only the elastic modulus and Poisson’s ratio are given. Corrected.
  • Assuming the materials are considered as linear elastic, the proposed approach does not consider the influence of the evolution of damage within the constituents. E.g. the failure of a material point does not influence its neighborhood. Thus, the proposed method consists of a simple (linear?) calculation with a subsequent calculation of the von-Mises stress and application of the maximum stress criterion. Yes, we are concentrated on the safaty zone and th first crack(s) zone. The progressive gamage is outside this paper.
  • The developed “computer program” is not outlined in detail. The provided information is very broad. Since this a one of the main novelties of the article, the reviewer considers a more in detail discussion to be appropriate. Corrected. The „computer problem“ arises because we use two difficult programs: AMSYS for computation the local stresses (ANSYS is ucefull for this aim) and our C program for computations described in Section 3 (ANSYS has programming tools, but we found ANSYS APDL not wery suitable for us). It is the problem of computation systems integration, it is important for the problem under consideration, but it is a separate problem.

Round 2

Reviewer 1 Report

The paper can be accepted for publication after text editing. I attach the file with my corrections proposed.

Reviewer 2 Report

The reviewer recommends rejecting the article and resubmitting it after thoroughly revising the content, using experimental validation and real-world cases to validate the numerical models.

Reviewer 3 Report

The article was improved according to the reviewer's suggestion.

Reviewer 4 Report

The authors addressed the issues regarding typesetting and the English language. Concerning the content of the work, the reviewer still believes that the impact and interest is low. The authors use very simple linear elastic material laws which are not capable to reproduce the pronounced nonlinear material response of the polymer matrix. Thus, the mechanical response of the polymer matrix is overpredicted. Moreover, the composite is assumed to fail if one material point within either the fibers or matrix fails. This is a conservative criterion. It is known that a polymer-matrix composite is capable of carrying loads far beyond the first occurrence of microcracks. Combined with the overprediction caused by the simple linear elastic material law used for the matrix, a very conservative strength criterion is deduced. The question is raised whether this approach is of interest for the composite community, especially for the design of composite structures. In order to solve this issue, it is recommended to, at least, consider choosing a more realistic, nonlinear material law for the matrix material in order to solve the overprediction issue.
Finally, the reviewer does not feel comfortable to recommend the article for publication.